# Elucidation of the aberrant 3′ splice site selection by cancer-associated mutations on the U2AF1

Hisashi Yoshida[1], Sam-Yong Park[1], Gyosuke Sakashita[2], Yuko Nariai[2], Kanako Kuwasako[3], Yutaka Muto [3✉], Takeshi Urano[2] & Eiji Obayashi [2✉]

The accurate exclusion of introns by RNA splicing is critical for the production of mature mRNA. U2AF1 binds specifically to the 3′ splice site, which includes an essential AG dinu-cleotide. Even a single amino acid mutation of U2AF1 can cause serious disease such as certain cancers or myelodysplastic syndromes. Here, we describe the first crystal structures of wild-type and pathogenic mutant U2AF1 complexed with target RNA, revealing the mechanism of 3′ splice site selection, and how aberrant splicing results from clinically important mutations. Unexpected features of this mechanism may assist the future devel-opment of new treatments against diseases caused by splicing errors.

[1] Graduate School of Medical Life Science, Yokohama City University, 1-7-29 Suehiro-cho, Tsurumi-ku, Yokohama 230-0045, Japan. [2] Department of Biochemistry, Shimane University School of Medicine, 89-1 Enya-cho, Izumo 693-8501, Japan. [3] Faculty of Pharmacy and Research institute of Pharmaceutical Sciences, Musashino University, 1-1-20 Shin-machi, Nishitokyo-shi, Tokyo 202-8585, Japan. ✉email: ymuto@musashino-u.ac.jp; eijioba@med.shimane-u.ac.jp

**N**oncoding sequences known as introns are removed from precursor mRNA in the maturation process of mRNA. This reaction, RNA splicing, is well-controlled for gene regulation and for generating proteomic diversity[1,2]. The selection of the boundary between coding sequence (exon) and the intron is a crucial step for this regulation, and its misregulation underlies many human diseases[3–7]. U2AF (U2 snRNP auxiliary factor) was first identified as the protein complex that recruits U2 snRNP to the conserved branch region[8].

U2AF binds to the region around the 3′ end of the intron and the following coding sequence (exon)[9]. U2AF consists of large and small subunits[9]. U2AF large subunit, U2AF2, interacts both with the polypyrimidine tract in the intron and to SF3B1, a component of U2 snRNP, to form the A complex of the spliceosome[10,11]. On the other hand, U2AF small subunit, U2AF1, binds to the 3′ side of the boundary sequence between the exon and intron, known as the 3′ splice site (3′SS, shown in Fig. 1a)[12,13]. Then, the recognition of 3′SS by U2AF1 is a critical step for the determination of the excluded intron for the production of mature mRNA. If the correct 3′SS is not recognized by U2AF1, the following exon will be excluded from the mature mRNA. The 3′SS must be recognized properly for accurate splicing, and inappropriate binding of U2AF1 to other sites results in isoforms of the translated protein. Aberrant splicing due to misrecognition of the 3′SS is known to cause various human diseases[14–16]. Even single amino-acid mutations of U2AF1 may cause selection of cryptic or aberrant 3′SSs, leading to misregulation of alternative splicing[17–20]. Using genome-wide analysis, Ilagan et al., Kim et al., and Okeyo-Owuor et al. recently reported that the S34F/Y mutations of U2AF1 change the preferred 3′SS and enhance aberrant exon inclusion[17–19], leading to hematological malignancies, including myelodysplastic syndromes (MDS). Furthermore, very recently, Fei et al. and Esfahani et al. elucidated by genomic analysis and iCLIP-RNA sequencing that the S34F mutant of U2AF1 causes aberrant alternative splicing in lung adenocarcinomas[20,21]. In spite of its importance, the molecular mechanism of sequence-specific RNA recognition by U2AF1 has been poorly understood, especially from a structural point of view. Previously, we reported that

U2AF1 could bind to the RNA molecule on its own through the specific recognition of the AG dinucleotides and solved the ternary structure of the RNA-free form of U2AF1[22]. In this study, we solved the crystal structures of wild-type U2AF1 and pathogenic S34Y mutant U2AF1 complexed with RNA containing a 3′SS sequence, in order to clarify how U2AF1 recognizes target RNA accurately and how disease-related mutant of U2AF1 recognizes aberrant 3′SS.

## Results

**The overall structure of U2AF1 with RNA.** The amino-acid sequence of U2AF1 is highly conserved between human and fission yeast with 60% identity, except for RS domain (Supplementary Fig. 1). The amino acids involving in RNA binding and also pathogenic hot spot, Ser34 and Gln157 of human U2AF1, are all conserved in the fission yeast U2AF1, shown in Supplementary Fig. 1. The structure of yeast U2AF1 is therefore a promising model for the elucidation of mutation effects on human U2AF1. As U2AF1 was stabilized by the binding of U2AF2[23], structural and biochemical experiments were carried out with a complex of *Schizosaccharomyces pombe* U2AF1 with the short fragment of U2AF2 spanning residues 93–161 (the U2AF1 complex), as previously reported[22]. U2AF1 consists of conserved three domains, an N-terminal zinc finger (ZF1), a U2AF homology motif (UHM), and a C-terminal zinc finger (ZF2)[24]. In our previous crystal structure of the U2AF1 complex without RNA, the ZFs lie against the β-sheet of the UHM and contact each other, consistent with mutational experiments that suggested RNA might interact with both ZFs[22]. In order to clarify where and how U2AF1 binds to RNA, we solved the crystal structure of U2AF1 complex with RNA containing 3′SS sequence, 5′-UAGGU. The overall structure of the RNA-free protein is conserved in the RNA-bound U2AF1 complex[22], as shown in Fig. 1b and Supplementary Fig. 2a. RNA is bound by both ZFs, as shown in Fig. 1b. Nine copies of the U2AF1 complex with RNA are found in the asymmetric unit. The U2AF1 complex in all nine molecules has almost the same structure as that of RNA-free form of U2AF1 as shown in Supplementary Fig. 2a. Comparing the apo and complex forms yields a root-mean-square deviation (RMSD) values for different molecules in the asymmetric unit varying from 0.59 to 0.98 Å, and each copy of the complex shows the same RNA contacts (Supplementary Fig. 2a, b).

Major structural differences between RNA-bound and RNA-free form are observed in the N-terminal region of U2AF1. Without RNA, N-terminal 14 amino-acid residues are disordered and not visible in the electron density. This region of U2AF1 is stabilized by interaction with RNA, and ten additional amino-acid residues (Leu5–Lys15) can be modeled in the RNA-bound form. The side chains of Glu12 and Lys15 interact with the 2′-hydroxyl group of the sugar portion of the guanine residue at −1 position (−1G) and the phosphate group between the uridine residue at −3 position (−3U) and the adenine residue at −2 position (−A), respectively. The Leu5–Tyr9 segment forms no contacts with the RNA, but the N-terminal region is stabilized in some cases by interaction with neighboring molecules in the crystal packing. In the present crystal structure, the first four bases of the 5′-UAGGU sequence (from −3U to the guanine at +1 position (+1 G) are held by the two ZF domains (Fig. 1b). The carbonyl oxygen of the uridine at the +2 position (+2U) is found to form a hydrogen bond with the guanidyl group pf Arg150, which stacks against the guanine base of +1 G. Various orientations of the +2U base were seen among the different copies in the asymmetric unit, as shown in Supplementary Fig. 3, and it is probable that the nucleotide sequence at +2 position is not strictly recognized by U2AF1.

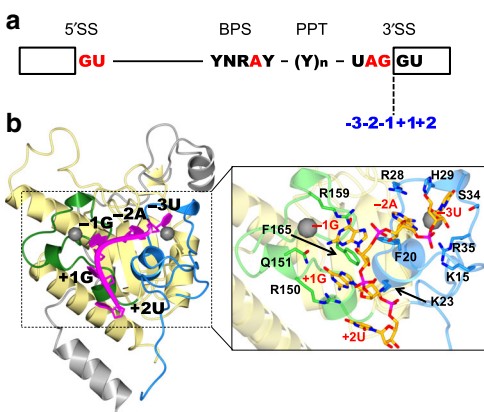

**Fig. 1 Structure of the U2AF1 complex bound to 3′ splice site RNA. a** Sequence elements required for splicing. Boxes indicate exons; Y pyrimidine, R purine, N any nucleotide, 5′SS 5′ splice site, BPS branch-point sequence, PPT polypyrimidine tract, 3′SS 3′ splice site. The dotted line indicates the exon boundary at the 3′ splice site, and numbers indicate the distance from the boundary. **b** Crystal structure of U2AF1 complexed with RNA, 5′-UAGGU. The N-terminal zinc finger (2-43, ZF1), U2AF homology motif (UHM, 44-141), C-terminal zinc finger (ZF2, 143-170), U2AF2 fragment (105-159), and RNA are colored blue, yellow, green, gray, and magenta, respectively. Inset: close-up view of the RNA-binding region in U2AF1. RNA is shown in stick representation colored with orange.

Upon binding to RNA, some amino-acid residues, especially in the ZF domains, change their side-chain configuration (Supplementary Fig. 2c, d), so that the RNA interactions with ZF1 are different from the model based on the structure of RNA-free U2AF1[22], and the RNA-bound form of canonical zinc fingers (Supplementary Fig. 4). The aromatic rings of Phe20 and Phe165 are rotated slightly to stack with −2A and +1G, respectively (Supplementary Fig. 2c, d). Phe165 also forms hydrophobic contacts with the −1G base. Arg28, which hydrogen bonds to Asn164 in the free form[22], is pulled toward the RNA molecule so that its guanidyl group stacks against the −2A base, and interacts with the 2′ hydroxyl group of −3U (Supplementary Fig. 2c). The side chain of Asn164 is displaced away from the RNA.

**Recognition of AG dinucleotides**. The experimental structure of U2AF1 complexed with RNA clarifies how the AG dinucleotide in the 3′SS sequence is recognized accurately by U2AF1. Mutation of the AG nucleotide in the 3′SS sequence causes a dramatic decrease in the binding affinity of U2AF1, but the basis of this sequence-specific interaction was not previously resolved[22]. As mentioned above, the −2A base is stacked by the aromatic ring of Phe20, and is sandwiched by the side chain of Arg28 in ZF1 (Fig. 2a and Supplementary Fig. 2c). Simultaneously, the guanidyl group of Arg28 could make a hydrogen bonding with the imidazole ring of His29 unexpectedly, holding the ring in place to recognize the −3U base (Supplementary Fig. 2c). This interaction was not observed in the crystal structure of U2AF1 complex without RNA.

The nearby Cys27, which coordinates a zinc ion (Fig. 2a), accepts a hydrogen bond from the N$^6$-amino group of the adenine −2A. Guanine has carbonyl oxygen at the corresponding 6th position and could not form an equivalent hydrogen bond, so Cys27 strongly prefers adenine at the −2 position in the RNA ligand.

The present wild-type crystal structure shows that the six-membered ring of −2A lies against the aromatic ring of Phe20 (Fig. 2a). Cytidine residue could possibly replace adenine as it has an N4-amino group (a donor of the hydrogen atom) and N3-nitrogen atom (an acceptor of the hydrogen atom). However, overlaying the cytosine ring with the six-membered ring of −2A, C1′ of the pyrimidine moiety then lies extremely close to the base of −1G (Supplementary Fig. 5). The spatial relationship between −2A and −1G is fixed by the positions of ZF1 and ZF2 (Fig. 1b). In addition, the O2 oxygen atom of the overlaid cytidine residue is also too close to the Oδ oxygen atom of Asn164. Adenine is therefore strongly favored at the −2 position site to prevent steric hindrance between the RNA backbone and the −1G base.

Furthermore, since the N6 atom of −2A is closely surrounded by Phe20, Cys27, Arg28, and Asn164, the methylation at this position of the base would cause significant steric repulsion (Fig. 3a). Compared with unmodified RNA (Fig. 4a and Table 1), ITC showed the affinity of U2AF1 for methylated RNA, 5′-UUm$^6$AGGU, is dramatically decreased, raising the $K_d$ to 53.8 μM (Fig. 3b). This clearly shows that m$^6$A modification strongly affects 3′SS selection by U2AF1. Although there is no reported clinical evidence, our data suggest that aberrant m$^6$A modification at 3′SS could block splicing at these positions[25,26].

On the other hand, the side chain of Phe165 (in ZF2) interacts with the guanine bases at positions −1 and +1 through perpendicular and parallel π–π stacking, respectively (Figs. 1b and 2b, c). The aromatic ring of Phe20 (ZF1) is also involved in the formation of the van der Waals contact surface with Phe165 for the base of −1G nucleotide (Fig. 1b). Thus, cooperative interaction between ZF1 and ZF2 plays an important role in the binding of the −1G nucleotide, which also lies against the guanidyl group of Arg159 (Fig. 2b). The O6-carbonyl oxygen of −1G interacts with the side-chain amide group and with the main-chain nitrogen of Asn164 (Fig. 2b). Cysteine residues Cys163 and Cys149 form hydrogen bonds with the N1 and N2 atoms of guanine −1G (Fig. 2b), respectively. These hydrogen bonds greatly increase the A/G discrimination, selecting guanine at the −1 position over adenine. This is consistent with our previous result that mutations at −1G affect the interaction between RNA and U2AF1 more strongly than at −2A[22]. Taken together, our results show both ZFs play a cooperative role in the RNA-binding site, particularly in the accurate recognition of the AG dinucleotide sequence in the 3′SS sequence through direct interactions.

**Nucleotide at −3 position and pathogenic mutants S34F/Y**. It has been reported that patients with cancers or MDS have mutations in some splicing factors, especially proteins involved in intron recognition[15,27,28]. One of the hot spot for such mutations is Ser34 in U2AF1; replacement of Ser34 with Tyr or Phe residues induces aberrant splicing[29]. Usually, the −3 position of 3′SS is occupied by pyrimidine[30], and SELEX confirms that the U2AF complex prefers a pyrimidine base at this position[12]. However, whole-exome sequence analysis by Ilagan et al., Kim et al., and Okeyo-Owuor et al. showed that A or C are found much more frequently at the −3 position of the 3′SS ((A/C)AG) recently, if Ser34 is replaced by Phe or Tyr in hematological malignancies[17–19]. In addition, Fei et al. and Esfahani et al. reported that in lung adenocarcinomas, the U2AF1 S34F mutant preferentially binds to CAG at 3′SS, unlike wild-type U2AF1[20,21]. These results suggested that while the wild-type protein requires −3U to function efficiently, these pathogenic mutants of U2AF1 can accept A or C at this position. In order to confirm the sequence preference at −3 position, we measured the binding affinities of U2AF1 to several RNA molecules with a variety of

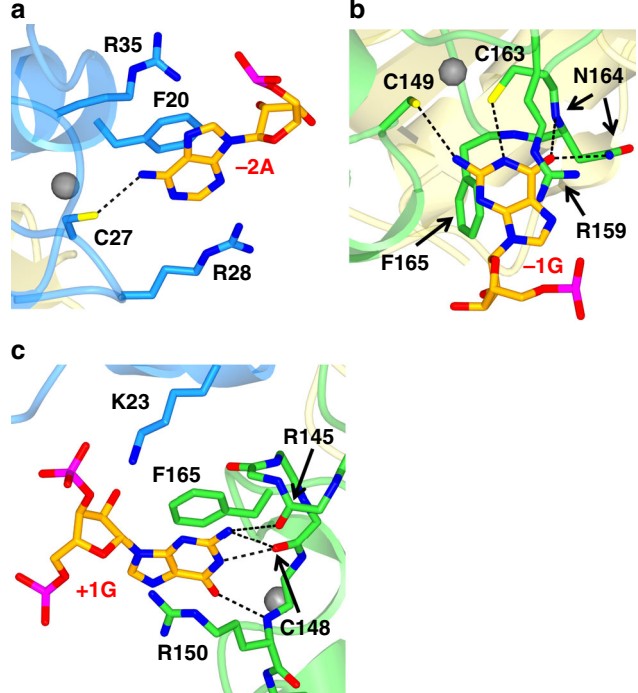

**Fig. 2 Representative views of the U2AF1 interactions with each nucleotide at the 3′ splice site. a** Interaction of U2AF1 with adenine at −2 position. **b** Interaction of U2AF1 with guanine at −1 position and (**c**) with guanine at +1 position structures. In U2AF1, ZF1, ZF2, and RNA are colored as in Fig. 1b.

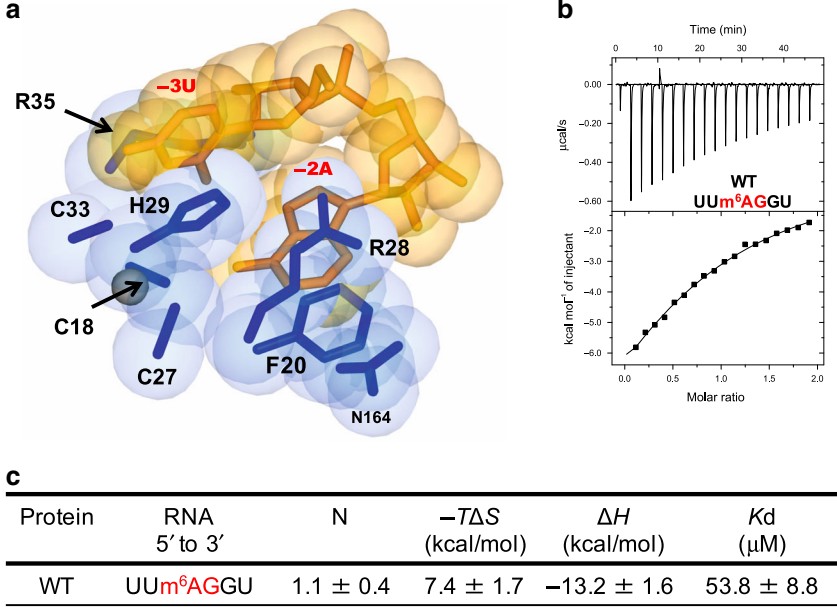

**Fig. 3 Specific interaction between the N⁶ amino group of −2A base and Cys27. a** Space-filling representation around the −2A base. The side chains of the U2AF1 amino-acid residues are shown in blue, and the RNA bases in orange. The interface around −2A base is intimately surrounded by the amino-acid residues of U2AF1. Thus, no space is found for the N⁶-methyl modification of the –2A base to interact with U2AF1. **b** Binding affinity of U2AF1 WT for the methyl modified RNA, 5′-UUm⁶AGGU, measured by ITC. Raw data and the corresponding binding curve are depicted. **c** RNA-binding activities calculated by ITC measurements. Mean value for the dissociation constant ($K_d$) with standard deviation is based on three independent measurements.

nucleotides at −3 position by isothermal titration calorimetry (ITC). We observed a $K_d$ value of 0.47 μM for six-nucleotide RNA, 5′-UUAGGU (Fig. 4a and Table 1). Replacement of uridine at the −3 position with adenine reduces the binding affinity 2.8-fold, to give a $K_d$ value of 1.7 μM (Fig. 4b and Table 1). On the contrary, the S34Y mutant binds to 5′-UUAGGU and 5′-UAAGGU with $K_d$ values of 0.27 μM and 0.46 μM, respectively (Fig. 4c and Table 1) and those for the S34F mutant were 0.37 μM and 0.39 μM, respectively (Table 1). The S34Y and S34F mutants therefore bind 5′-UAAGGU with an affinity similar to that of wild-type protein for 5′-UUAGGU. Furthermore, replacement of the uridine residue at the −3 position with a cytidine residue (5′-UCAGGU) weakens the binding to wild-type U2AF1, giving a $K_d$ value of 1.7 μM (Table 1). However, the S34F and S34Y mutants bind 5′-UCAGGU with $K_d$ values of 0.77 μM and 0.63 μM, respectively, similar values as that for 5′-UUAGGU (Table 1). These ITC results are consistent with the previous analysis of lung adenocarcinomas that S34F mutant prefers 3′SS sequences, including a CAG motif[20,21]. In contrast to wild-type U2AF1, therefore, the S34Y and S34F mutants show little discrimination for the base at −3 position, except for rejecting guanosine. In order to understand the structural basis for these binding preferences, the crystal structures of U2AF1 complexed with 5′-UAGGU or 5′-AAGGU were solved and compared.

In the 5′-UAGGU complex, the uracil base of −3U is stacked with the imidazole ring of His29 and surrounded by Ser34, Arg35, and a short helix of ZF1 (Fig. 4d). The stacking interaction between the imidazole and the uridine rings is crucial for the binding of RNA to U2AF1. ITC measurements revealed that the replacement of histidine with alanine at this position decreases the RNA-binding affinity significantly (Supplementary Fig. 6). The O4-carbonyl oxygen of −3U forms a hydrogen bond with the main-chain amide proton of Ser34, whose hydroxyl side-chain hydrogen bonds to N3 imino proton of –3U (Fig. 4d). The N3 imino proton of −3U also interacts with the sulfur atom of Cys33 (Fig. 4d). If −3U is replaced with cytidine, then the hydrogen bond to Ser34 would be lost as cytidine does not have a

hydrogen-bond acceptor atom at the N4 position. In addition, the histidine residue prefers uracil base to cytosine base in the protein–RNA interaction as reported previously[31], which could be true for the recognition of U2AF1 for the −3 position.

The model of the 5′-UAGGU complex also shows a hydrogen bond between the O2-carbonyl oxygen of –3U and N6 amino-proton of −2A (Supplementary Fig. 7a). In contrast, the complex structure with 5′-AAGGU shows that adenine at −3 position (−3A) is unable to make this interaction, and forms only one hydrogen bond, with the main-chain carbonyl oxygen of Arg32 (Fig. 4e and Supplementary Fig. 7b). In the wild-type protein, the pocket accepting the RNA base at −3 position is too shallow to accommodate adenine base, and the loss of hydrogen bonds with the ligand is consistent with RNA-binding affinities determined by ITC.

To elucidate further the impaired RNA recognition mechanism of the S34F/Y mutants, we attempted to crystallize the S34Y mutant with bound RNA (5′-UAGGU, 5′-AAGGU, or 5′-CAGGU). This mutant was chosen because its solubility is higher than that of S34F mutant, making it more likely to find suitable crystallization conditions. Although the 5′-AAGGU and 5′-CAGGU complexes did not crystallize, the crystal structure of the complex between the S34Y mutant and 5′-UAGGU was determined. In the complex, O4-carbonyl oxygen of −3U maintains a hydrogen bond with the amide proton of the mutated residue (Tyr34), despite a small but distinct position shift of the uridine base compared to that in wild-type U2AF1 (Fig. 4f). Along with the imidazole ring of His29, the aromatic ring of Tyr34 also stacks against the −3U base, forming a π–π interaction that is absent from the wild-type complex (Fig. 4f). These are several known cases in which Phe and Tyr side chains interact with RNA bases, including cytosine[31]. This additional π–π interaction is strong enough to allow the pathogenic S34Y mutant to select AAG or CAG sequences as a 3′SS.

In order to confirm changes in splice site selection in the human cell, we have performed splicing assays using the *ATR* minigene (Fig. 5). For all of the U2AF1 constructs (WT, S34F,

| Protein | RNA 5′ to 3′ | N | $-T\Delta S$ (kcal/mol) | $\Delta H$ (kcal/mol) | $K_d$ (μM) |
|---------|--------------|-----|--------------------------|------------------------|-------------|
| WT | UUm⁶AGGU | 1.1 ± 0.4 | 7.4 ± 1.7 | −13.2 ± 1.6 | 53.8 ± 8.8 |

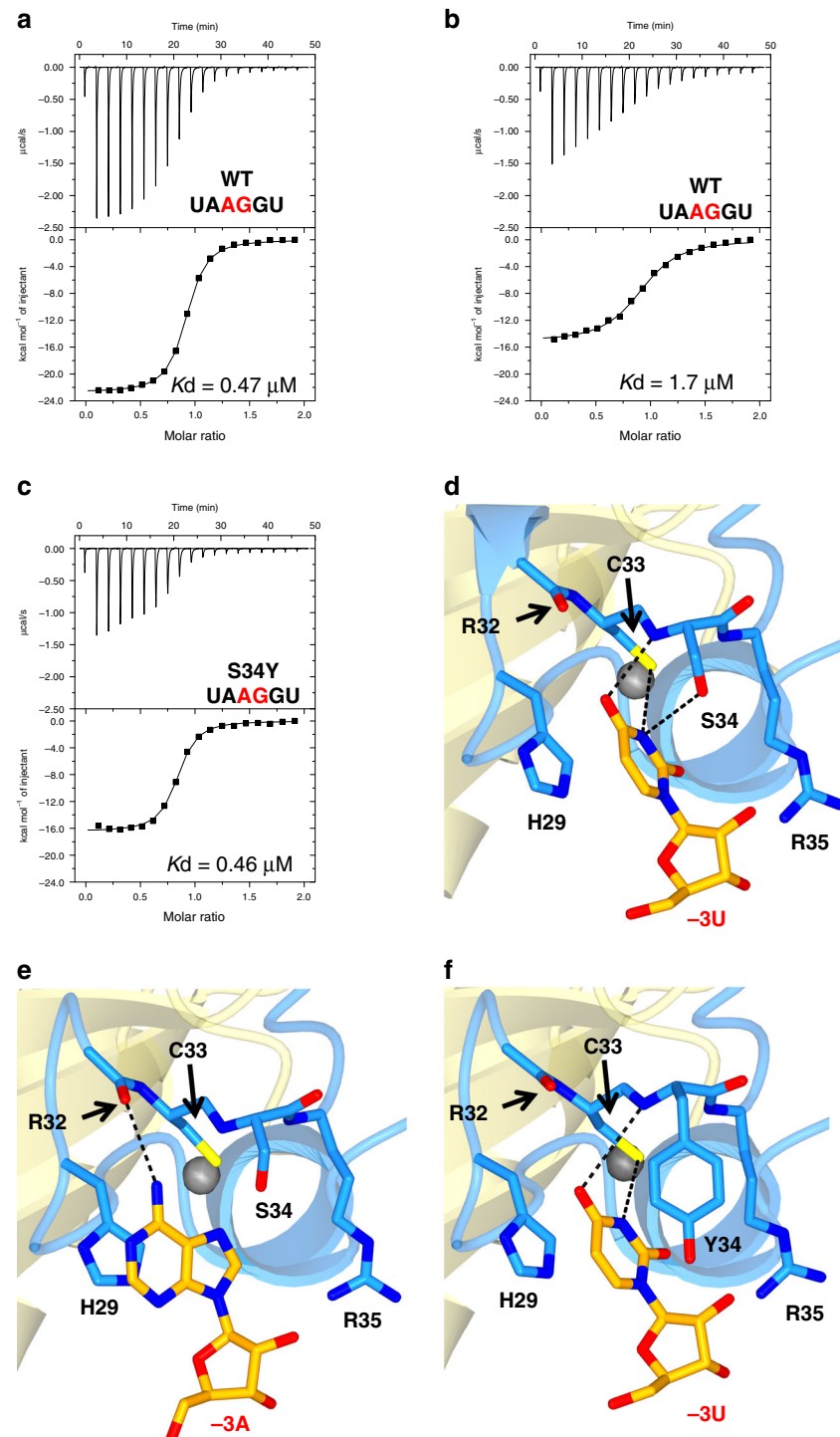

**Fig. 4 Interaction of U2AF1 with the nucleotide at −3 position. a** Binding affinities of U2AF1 WT for the 5′-UUAGGU RNA, **b** for the 5′-UAAGGU RNA, and **c** S34Y for the 5′-UAAGGU RNA measured by ITC. Raw data and the corresponding binding curve are depicted. Mean value for the dissociation constant ($K_d$) with standard deviation is based on three independent measurements. Representative views of the interaction (**d**) between U2AF1 WT and −3 uridine, **e** between U2AF1 WT and −3 adenine, and (**f**) between U2AF1 S34Y and −3 uridine. U2AF1 and RNA are colored as in Fig. 1b.

and S34Y), the exon inclusion mainly occurred at 3′SS with sequences TAG and CAG. WT U2AF1 showed almost no splicing activity at the AAG site, but the S34F and S34Y mutants both showed 30% of total activity at such 3′SS (Fig. 5b, c), in agreement with earlier studies, while both WT and mutants showed high splicing activity for the 3′SS with CAG sequence. The S34F/Y mutations clearly allow much greater flexibility in the selection of splice sites with U, C, or A at the −3 position. It implied that

some chemical compound that specifically binds to the space formed by His29 and Tyr34/Phe34 could affect the abnormal cells with the mutations.

**Exon sequence at +1 position and pathogenic mutants Q157P/R.** Our present crystal structure indicates that U2AF1 has a strong preference for guanine at the +1 position in the exon sequence. +1 G forms a π–π interaction with Phe165 and

**Table 1 RNA-binding activities to the U2AF1 evaluated by ITC measurement.**

| RNA 5' to 3' | $K_d$ (µM) WT | S34F | S34Y |
|---|---|---|---|
| UAAGAU | 19.3 ± 3.5 | 3.4 ± 0.8 | 3.7 ± 2.0 |
| UAAGCU | 13.5 ± 1.6 | 2.1 ± 0.2 | 1.9 ± 0.6 |
| UAAGGU | 1.7 ± 0.2 | 0.39 ± 0.05 | 0.46 ± 0.05 |
| UAAGUU | 25.9 ± 0.9 | 4.1 ± 0.4 | 3.0 ± 1.1 |
| UCAGAU | 12.1 ± 0.8 | 4.9 ± 0.4 | 4.2 ± 1.6 |
| UCAGCU | 10.1 ± 2.0 | 6.1 ± 1.3 | 3.6 ± 1.4 |
| UCAGGU | 1.7 ± 0.2 | 0.77 ± 0.16 | 0.63 ± 0.10 |
| UCAGUU | 16.4 ± 2.0 | 7.7 ± 0.5 | 4.9 ± 1.8 |
| UGAGAU | 63.0 ± 17.7 | 33.7 ± 11.9 | 38.3 ± 3.1 |
| UGAGCU | 35.6 ± 5.6 | 15.2 ± 4.4 | 29.3 ± 11.6 |
| UGAGGU | ND | ND | ND |
| UGAGUU | 60.7 ± 24.7 | 32.8 ± 4.5 | 38.9 ± 7.9 |
| UUAGAU | 2.7 ± 1.1 | 2.3 ± 0.4 | 3.0 ± 2.9 |
| UUAGCU | 3.6 ± 0.4 | 1.5 ± 0.1 | 1.9 ± 1.0 |
| UUAGGU | 0.47 ± 0.03 | 0.37 ± 0.09 | 0.27 ± 0.03 |
| UUAGUU | 6.8 ± 0.7 | 3.1 ± 0.3 | 1.3 ± 0.2 |

All measurements were performed in triplicate independently.
Errors (±) are standard deviation.
Binding parameters of all measurements are shown in Supplementary Table 1 and 2, and raw data are shown in Supplementary Fig. 12.
The nucleotides colored by red indicates the conserved 3′ splice site sequence.

cation–π interaction with Arg150 involves the six-membered ring of the base, which lies further from the RNA backbone than the five-membered ring (Fig. 2c). Pyrimidine (C or U) bases at this position would not be large enough to interact with Arg150 or Phe165. +1 G also interacts directly with the amino group of −1G (Supplementary Fig. 7c) and the main-chain carbonyl oxygens of Arg145, Glu146, and Cys148 through hydrogen bonds (Fig. 2c). Placing an adenine residue at this position would remove all these hydrogen bonds, explaining why U2AF1-binding affinity for RNA sequences with guanine at the +1 position is highest, whatever the preceding intron sequences (Table 1). Our model is therefore consistent with earlier SELEX experiments showing that guanine is the preferred nucleotide at this position for U2AF1[12].

Another mutation hot spot in human U2AF1 is found at Q157, which corresponds to yeast Q151 (Supplementary Fig. 1). This residue is located close to the −1G and +1 G bases in the wild-type U2AF1 complex structure (Fig. 1b), and it is likely that mutations at this position could affect RNA recognition. It was reported by Iligan et al. that exon recognition was suppressed by the mutation Q157R mutant when adenine was located at the +1 position. Furthermore, in our previous study, the Q151R mutant of yeast U2AF1 was found to bind to UAG-containing 3′SS more tightly than WT U2AF1, and show different preferences for the −1 position[22]. For example, the WT U2AF1 cannot recognize 3′ SS with UAU, but the Q151R mutant does. Therefore, our present complex structure explains the importance of Q157 (Q151 in yeast) for the fidelity of the recognition of the bases at −1 and +1 positions.

**Comparison of RNA recognition by other CCCH-type ZFs.** To date, several protein–RNA complex structures have been solved for members of RNA-binding CCCH-type zinc finger domains (ZFs). Based on overlays of these previously determined complex structures (MBNL1 ZF2 and 3, TISIId ZF1, mouse Unkempt proteins ZF3 and 6, CPSF30 ZF2 and 3, and Nab2 ZF5)[32–37], we could identify four putative RNA-base recognition sites, pockets A, B, C, and D with crucial amino-acid residues at six key positions (i–vi) on the CCCH domains (Supplementary Fig. 8).

Basically, most CCCH-type ZF domains have the pocket B formed by conserved Phe or Tyr residues at the (vi) position and Arg or Lys residues at the (i) position, and have the pocket C formed by Lys, Arg or aromatic residues at the (iv) position and the aromatic amino-acid residue at the (vi) position (Supplementary Fig. 8)[32,34].

The TISIId and Unkempt ZFs have an additional pocket A (Supplementary Fig. 8). In these models, the bound RNA is aligned on ZF so that a base near the 3′ end is located in pocket C. These members could be classified in Group I (Supplementary Fig. 9). On the other hand, several ZFs do not have pocket A, and these can be classified into two groups, Group II and III, according to the absence or presence of an aromatic amino-acid residue at the (ii) position. In members of Group II, including U2AF35 ZF2, and MBNL1 ZF2 and ZF3 (Supplementary Fig. 9), one RNA base is accommodated in pockets B, and an RNA base near the 5′-end is located at pocket C. Therefore, whether pocket A is utilized or not (Group I and Group II) seems to be important for the correct recognition of RNA bases.

On the other hand, ZF5 of Nab2 protein and CPSF40 ZF2 and 3, in Group III, have aromatic amino-acid residues at the (ii) and (vi) positions (Supplementary Fig. 9). In these ZFs, an RNA-base stacks with the aromatic ring at (ii) position and another base in pocket D. U2AF1 ZF1 is also a member of Group III. However, as mentioned previously, U2AF1 ZF1 has an Arg residue at the (vi) position[22]. Intriguingly, RNA bases are located at pockets C and D in U2AF1 ZF1, instead of pockets B and D as in Nab2 ZF. In the Pfam32.0 database[38], this combination of the amino-acid residues at (iv) and (vi) positions is rare in the zf-CCCH1 family and appeared in the U2AF1 paralogs and PARP12 ZFs. In Group III, it is considered that the D position establishes the RNA alignment by accepting the nucleotide at the 3′-end. Taken together, current evidence suggests that pockets A and D play an important role in the disposition of the RNA bases on ZF domains.

**Discussion**
This study has clarified the molecular mechanism of the recognition of 3′SS by U2AF1 in an early splicing step. Recently, many structures of spliceosomal complexes at the different stages in the splicing reaction were elucidated by cryo-electron microscopy[39–41]. These structures show that in the later spliceosomal complexes, the 3′SS is recognized through interactions with the branch site adenosine and with 5′SS by non-Watson Crick base paring[42–45]. However, in the pre-B complex, an early spliceosomal complex, the 3′SS could not be identified because the electron density showed disorder[46,47]. Although the A-complex structure of *Saccharomyces cerevisiae*, another early stage in spliceosome assembly, has been elucidated, the 3′SS is not seen at this stage of splicing in budding yeast, which has no homologous protein for U2AF1. Our present study is therefore the first report of

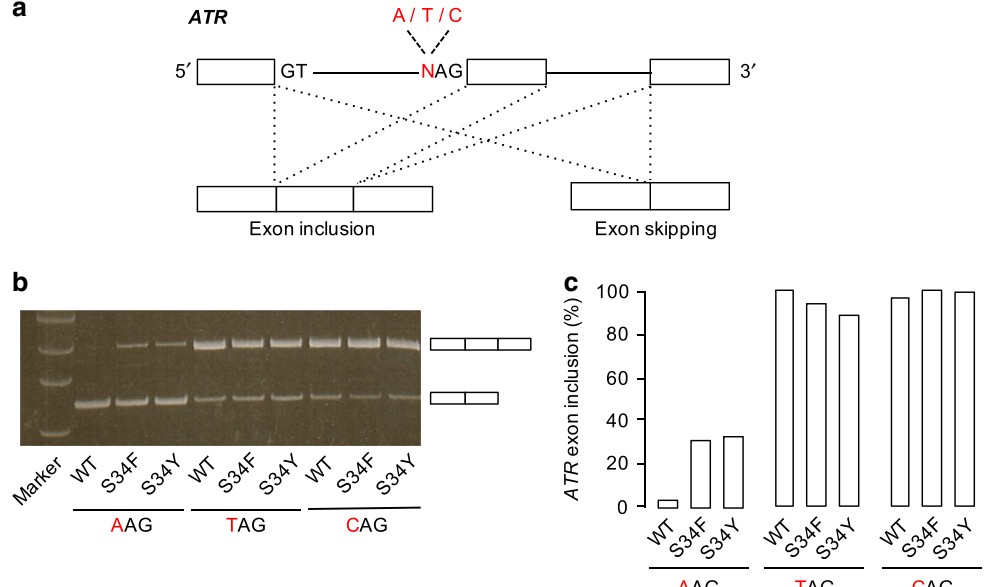

**Fig. 5 Minigene-splicing assay. a** Schematic diagram of ATR minigene designed for the splicing assay, which has different nucleotide A or T at the −3 position of the 3′ splice site. The minigene exons are shown as boxes, and introns as solid lines. Two different pre-mRNA splicing patterns are shown as exon inclusion and exon skipping. The RNA product of the exon inclusion is derived from the removal of two introns and junction of three exons, whereas that of the exon skipping is produced by the junction of exons at both ends. **b** Gel electrophoresis of RT-PCR products from HEK293 cells expressing WT or mutants of U2AF1. Upper fragments correspond to the product by the exon inclusion, and lower fragments correspond to that by the exon skipping. Raw image data of gel electrophoresis are shown in Supplementary Fig. 13. **c** The ratio of ATR exon inclusion and the skipping is quantified from the result of (**b**), and compared based on the amount from cells expressing WT with TAG minigene.

molecular details of how the 3′SS is recognized by spliceosomal proteins at an early splicing step.

Our study shows that the 3′SS AG dinucleotide is strongly recognized by the two ZFs of U2AF1 (Figs. 1b and 2a, b), and that $m^6A$ modification affects the 3′SS recognition by U2AF1 (Fig. 3b). Knockdown of the methyltransferase for $m^6A$, METTL3, or the demethylase for $m^6A$, FTO, is known to change the splicing pattern[48–50]. There is no evidence to date to indicate that U2AF1 interacts with $m^6A$, and our results suggest that accidental $m^6A$ modification of a 3′SS could block splicing. Further exome-wide analysis of RNA modification is required for a complete understanding of the relationship between $m^6A$ and alternative splicing.

Aberrant splicing of the exon inclusion at AAG 3′SS by S34F/Y mutants of U2AF1, as seen in MDS patients, is consistent with our structural analysis and RNA-binding affinity study. On the other hand, splicing errors at CAG 3′SS caused by S34F U2AF1, as seen not only in MDS but also in adenocarcinoma, cannot be explained by our results. Although the binding affinity of S34F mutant for CAG is higher than that of WT U2AF1, as well as the case for AAG, minigene assays showed that splicing efficiency at CAG 3′SS is almost the same for both the WT and mutant forms of U2AF1 (Fig. 5). Recently, Warnasooriya et al. reported that the mutation at Ser34 of U2AF1 influences the domain conformation of U2AF2, which affects the binding affinity of polypyrimidine tracts[51]. Therefore, it may be necessary to consider including other splicing factors working with U2AF1, U2AF2, and SF1.

Surprisingly, our models show the ZFs of U2AF1 also contribute to the preference for bases flanking the AG nucleotide (Figs. 2c and 4d), so that the RNA sequence most strongly bound is UAGG (Fig. 4a and Table 1). Pathogenic U2AF1 mutants have different sequence specificity from that of wild-type U2AF1 (Fig. 4b, c and Table 1). One of the most frequent pathogenic mutations, S34F/Y, is related to the unique base binding pocket of

ZF1 (Fig. 4f), so that targeting U2AF1 may be a useful approach for drug discovery to treat diseases caused by aberrant mRNA splicing[52–54]. Overall, our U2AF1 structure highlights how the intron is recognized early in the splicing process, and reveals how alternative splicing may arise due to specific mutations in U2AF1 associated with diseases, such as MDS and cancer.

## Methods

**Sample preparation and crystallization.** RNAs for the crystallization and ITC measurements were purchased from FASMAC (Kanagawa, Japan).

Expression and purification of yeast U2AF1 complexed with the short fragment of U2AF2 (93–161) were performed as previously described[22]. For co-crystallization of U2AF1 with RNA, protein–RNA complex was produced by the addition of RNA to concentrated protein in a stoichiometric molar ratio of 1:1.5. Crystals were obtained by the hanging-drop vapor-diffusion technique. In total, 20 mg/ml of protein–RNA complex in 20 mM Tris-HCl (pH 8.0), 100 mM NaCl, and 1 mM TCEP was mixed with an equal amount of reservoir solution 10% (v/v) PEG35000, 0.1 M sodium citrate (pH 6.0) and equilibrated against reservoir solution at 4 or 20 °C. Cryo-protection was achieved in several steps by in-well buffer exchange with increased ethylene glycol concentration from 0 to 20%. Cryo-protected crystals were flash-frozen in liquid nitrogen for data collection.

**Structure determination and refinement.** Diffraction data were collected on beamline 17A at Photon Factory, and on beamline NE3A and NW12A at Photon Factory Advanced Ring in Tsukuba, Japan. All data were processed and scaled using XDS and Aimless[55–57]. The space group was found to be $P2_1$, with nine molecules in the asymmetric unit. The structures were solved by molecular replacement using Molrep[56], with the U2AF23 structure (Protein Data Bank entry 4YH8)[22] as the search model. The map was of good quality, allowing RNA of the model to be traced readily (Supplementary Fig. 10). Manual model building was performed using COOT[58], and refinement was carried out with Phenix-refine[59]. Non-crystallographic symmetry restraints were applied to each chain. Validation of the final model was carried out using MolProbity[60]. The Ramachandran statistics for WT-UUAGGU complex were 96.44% favored with no outliers, for WT-UAAGGU complex were 96.37% favored with no outliers, and for S34Y-UUAGGU complex were 95.87% favored with no outliers. A summary of the data collection and refinement statistics is given in Table 2. Atomic coordinates and structure factors of the complex have been deposited in the Protein Data Bank with accession code 7C06 for WT-UUAGGU, 7C07 for WT-UAAGGU and 7C08

## Table 2 Data collection and refinement statistics.

| | WT-UUAGGU (PDB: 7C06) | WT-UAAGGU (PDB: 7C07) | S34Y-UUAGGU (PDB: 7C08) |
|---|---|---|---|
| *Data collection* | | | |
| Space group | $P2_1$ | $P2_1$ | $P2_1$ |
| Cell dimensions | | | |
| *a, b, c* (Å) | 93.8, 255.1, 94.1 | 94.5, 257.0, 94.6 | 94.5, 259.0, 94.8 |
| α, β, γ (°) | 90, 101.1, 90 | 90, 100.8, 90 | 90, 100.7, 90 |
| Resolution (Å) | 48.86–3.02 | 48.17–3.20 | 48.40–3.35 |
| | (3.08–3.02)* | (3.27–3.20)* | (3.43–3.35)* |
| $R_{merge}$ | 0.113 (1.040)* | 0.117 (0.802)* | 0.230 (1.158)* |
| $R_{pim}$ | 0.046 (0.434)* | 0.058 (0.404)* | 0.094 (0.466)* |
| $CC_{1/2}$ | 0.997 (0.727)* | 0.996 (0.638)* | 0.971 (0.742)* |
| $I / \sigma I$ | 17.0 (2.1)* | 10.2 (2.1)* | 8.3 (2.2)* |
| Completeness (%) | 100.0 (100.0)* | 100.0 (100.0)* | 100.0 (100.0)* |
| Redundancy | 7.1 (6.7)* | 4.9 (4.8)* | 7.0 (7.1)* |
| *Refinement* | | | |
| Resolution (Å) | 3.02 | 3.20 | 3.35 |
| No. of reflections | 84,653 | 72,745 | 64,508 |
| $R_{work}/R_{free}$ | 0.2146/0.2487 | 0.2547/0.2854 | 0.2143/0.2523 |
| No. of atoms | | | |
| Protein | 17,455 | 17,409 | 17,486 |
| RNA | 945 | 963 | 945 |
| Zn | 18 | 18 | 18 |
| *B*-factors | | | |
| Protein | 73.2 | 79.8 | 72.3 |
| RNA | 81.9 | 92.1 | 80.9 |
| Zn | 57.9 | 60.6 | 48.9 |
| R.m.s. deviations | | | |
| Bond lengths (Å) | 0.002 | 0.002 | 0.002 |
| Bond angles (°) | 0.511 | 0.447 | 0.500 |

*Values in parentheses are for the highest-resolution shell.

for S34Y-UUAGGU. Structural figures were rendered for chain-A, B, and C using CCP4MG and MOLMOL[61,62]. Protein–RNA interactions were considered as hydrogen bonds between suitable atoms 2.3–3.5 Å apart, or hydrophobic interactions between apolar atoms 3.5–3.9 Å apart. The analysis was carried out with COOT and LIGPLOT[63] (Supplementary Fig. 11).

**ITC experiments**. All calorimetric titrations were carried out on iTC200 calorimeters (Malvern Panalytical). Protein samples were dialyzed against the buffer containing 20 mM HEPES (pH 7.0) and 100 mM NaCl. The sample cell was filled with a 50 μM solution of RNA, and the injection syringe was filled with 500 μM titrating U2AF complex wild type or mutants. All experiments typically consisted of a preliminary 0.4-μL injection followed by 18 subsequent 2-μL injections every 150 s. All of the experiments were performed at 25 °C. Data for the preliminary injection, which were affected by diffusion of the solution from and into the injection syringe during the initial equilibration period, were discarded. Binding isotherms were generated by plotting heats of reaction normalized by the moles of injected protein versus the ratio of the total injected one to the total RNA per injection. The heat of infections was corrected by subtracting the heat of dilution of the protein into the substrate-free buffer. The data were analyzed using Origin software with a single site-binding model. The dissociation constant values are summarized in Table 1, other binding activities are shown in Supplementary Table 1 and Supplementary Table 2, and raw data are shown in Supplementary Fig. 12.

**Minigene-splicing assay**. Inserts containing the *ATR* genomic locus (Chr 3: 142168271-142172075, deleting the region 142169744-142171669) with a single mutation at −3 position of the 3′ splice site were chemically synthesized by Fasmac (Kanagawa, Japan) and were cloned into pcDNA vector. The coding sequence of human U2AF1 was amplified by PCR using cDNA and was cloned into pcDNA3 vector with mCherry tag. The single mutation for S34F and S34Y of human U2AF1 was generated by PCR. Both minigene and U2AF1 expression plasmid vectors were injected into HEK293 cells, and infected cells were selected and collected by FACSAria II (BD Bioscience). After lysis of cells, spliced minigene was checked by RT-PCR using primers, 5′-AAGCTTCCACCATGGTCTTAAAG-3′ and 5′-GTCGCTGCTCAATGTCAAGA-3′. For quantitative analysis, the gel image was acquired using ImageQuant LAS 4000 (GE Healthcare) and quantified using Multi Gauge software (Fujifilm, Japan). Two replicas were performed for each experiment.

**Reporting summary**. Further information on research design is available in the Nature Research Reporting Summary linked to this article.

## Data availability
Atomic coordinates and structure factors of the complex have been deposited in the Protein Data Bank with accession code 7C06 for WT-UUAGGU, 7C07 for WT-UAAGGU, and 7C08 for S34Y-UUAGGU. All data are available from the corresponding author upon reasonable request.

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

## Acknowledgements

We thank the staff at the Photon Factory beam-lines for assistance with data collection. We also thank Professor Jeremy Tame for critical reading of the paper. We would like to acknowledge the technical expertise of the Center for Integrated Research in Science, Shimane University. This work was supported by JSPS KAKENHI Grant Number 18K06086 and JP19H05779 from the Ministry of Education, Culture, Sports, Science and Technology of Japan, and by grants from Musashino University Gakuin Tokubetsu Kenkyuhi to Y.M. This research was also supported by Platform Project for Supporting Drug Discovery and Life Science Research (Basis for Supporting Innovative Drug Discovery and Life Science Research (BINDS)) from AMED under Grant Number JP17am0101001 and JP18am0101076.

## Author contributions

H.Y., T.U., Y.M., and E.O. conceived and designed the project. H.Y., S.-Y.P., and E.O. carried out the crystallographic study. H.Y. and E.O. analyzed the binding affinity of protein and RNA using ITC. G.S., Y.N., and T.U. performed minigene-splicing assay. T.U., Y.M., and E.O. wrote the paper, and H.Y., K.K., Y.M., and E.O. discussed the results and conclusions.

## Competing interests

The authors declare no competing interests.
