## [Peer Review File · Nature Communications]

Reviewers' comments:

Reviewer #1 (Remarks to the Author):

Yoshida et al. present the crystal structures of the fission yeast U2AF1 splicing factor subunit bound to the heterodimerization domain of U2AF2 and to RNA sites, and also of its myelodysplasia-associated S34Y mutant. The structures are potentially important since to date, no high resolution structures of the U2AF1 subunit recognizing the splice site are available. However, technical concerns as well as concerns with the presentation dampens the enthusiasm of this reviewer.

Major comments:

1) The studies focus on the fission yeast homolog rather than the disease-relevant human homolog. How similar are the proteins? Please justify extrapolating the results from the yeast homolog to interpretations of the human disease.

I also feel that the use of the yeast protein should be more clear and honestly disclosed in the manuscript text.

2) The S34Y mutation is minor in MDS compared with the major S34F mutation. Please clarify why the S34Y mutation chosen as the focus of structure determination, how common it is in MDS, and how an S34F mutation might be similar in effects or differ.

3) The Q157P/R mutations are the next most common in MDS after S34F. Please describe the Q157 homologous residue of the fission yeast protein. How do the Q157 mutations affect splicing and is it supported by RNA interactions? This information is available from the structure but does not seem to have been analyzed.

4) The S34F mutation of the human protein influences -3C vs -3U choice rather than -3Py vs -3A. The focus on the -3A for structural biophysics seems misplaced or peculiar to the yeast protein. Can the structures give insight into why the human protein splices -3C and skips -3U?

5) The resolution of the structures does not appear sufficient to resolve hydrogen bonds since the Rmerge is >1 in the highest resolution shell. Moreover, the highest reported resolution is 3.35 for the S34Y structure. Please report R_{pim}. To what resolution is the R_{pim} <50% for the three structures, i.e. reliable?

Especially since only five main text figures are planned and the focus of the paper is crystal structures, the experimental data table would be more appropriate given in the main text, for full disclosure of the limits of the structure quality to the reader.

6) Please state the coordinate error and if justified by the structure quality report hydrogen bond

distances. If reporting hydrogen bond distances cannot be justified, please say so for the non-structural biologist audience.

In a few cases, the interactions appear to be mis-interpreted. For example, cysteines that are liganded to zincs are said to be involved in hydrogen bonds. This is highly unlikely, especially since cysteine side chains are hydrophobic in character. I suggest rephrasing these statements.

Ligplot could be used to plot the interactions in an unbiased manner.

7) Please give an unbiased electron density map of the RNA interactions. The 2Fo-Fc map shown in Supplementary data is a biased map and should not be used for publication.

8) Table of ITC data are given, but the data lacks errors and statistical significance. Were the experiments replicated? If not, the data should be replicated before publication. Please provide standard deviations of replicates and t-tests. Also, ALL isotherms should be shown to support the quality of the data.

9) Appropriate literature in the field is mis-cited (or not cited). Examples include but are not limited to:

“misregulation underlies many human diseases 3-7.” Has mix of primary lit and reviews and all hematologic malignancies not “many human diseases”.

The reference for U2AF binding the intron is a review, which I don't believe even mentions U2AF.

“U2AF2, interacts both with the polypyrimidine tract in the intron and to SF3B1”

Reference 11 does not have anything to do with SF3b binding U2AF. Reference 1 is a review.

U2AF1 contacts 3'SS – 13 & 14 are wrong, it should be Green Nature and Valcarcel MCB

Aberrant splicing due to mis recognition of 3pSS .. “and so on” – please elaborate.

References are reviews, could be primary for MDS, only two major ones Yoshida and Graubert

Ilagan and Kim S34F/Y made splicing program “confused” to enhance exon inclusion

Should be Graubert Leukemia and Varmus Plos Genetics as well as Ilagan

Not just exon inclusion – exon skipping, cryptic 3' SS, alternative 3'SS also occur

10) There are only a few structures of CCCH zinc knuckles bound to RNA, including this one. Could the authors please compare the RNA orientations between the two U2AF1 zinc knuckles and also with the other available high resolution structures of zinc knuckles (Unkempt and Yth1 as well as the two cited herein, possibly others). Are there common themes? (an aromatic-arg packing is mentioned but this reviewer is surprised because by memory I think the available structures are very different from one another with regard to orientation of the bound RNA). This information would be useful to the field of

RNA binding proteins

Minor comments:

- 11) U2AF1 does not have an RRM; It has a U2AF Homology Motif (UHM)
- 12) “crystal lattice” – not due to “lattice”, should be due to “packing”
- 13) How many atoms of what type contribute to the reported RMSDs
- 14) Please use prime rather than single quote for 3' or 5' of RNA

Reviewer #2 (Remarks to the Author):

U2AF1 is important for accurate 3' ss recognition and U2AF1 mutations are frequently observed in myelodysplastic syndrome or cancers. In this paper, the authors reported the first crystal structures of the WT and pathogenic mutant U2AF1 in complex with RNA containing the 3' ss. These structures revealed the mechanism of 3' ss selection by U2AF1 and how mutant U2AF1 changes 3' ss preference. The paper provided important insights in 3' ss recognition and disease mechanism of mutant U2AF1, but also has multiple issues that need to be addressed.

1. Page 7. There are multiple places where Fig. 4a was cited while Fig. 3a seems to be the correct figure.

2. Page 7. The authors stated that “It was believed that Phe20 only contacted the distal region of the base, preventing the steric hindrance with the sugar backbone, and favouring purine over smaller pyrimidine base at this site.”

Who believed? And what is evidence that support this belief?

3. Page 7. I am not convinced that the authors' argument that Cys27 strictly discriminates the adenine base is sound. The authors stated that Cys27 forms a hydrogen bond with the N6-amino group of -2A. If A were to be changed to G, the corresponding 6th position will be a carbonyl oxygen which will work as a proton acceptor and cannot form the equivalent hydrogen bond. However, why can't the sulfur atom of Cys27 serve as a hydrogen bond donor in this case to still form a hydrogen bond with a G nucleotide at -2 position? In fact, there are literature (for example PMID 19089987) reporting that the sulfur atom of a Cys is a poor hydrogen bond acceptor, but moderately good donor.

4. Page 8 and Fig. 3b.

The authors stated that the Kd is 46.9uM. The binding was never saturated and an accurate Kd cannot be derived from the data the authors presented here.

5. Page 8. The authors talked about Arg164. Shouldn't residue 164 be an Asn according to Fig. 2b?

6. At the beginning of page 10, the authors stated that "These results suggested that while the wild-type protein requires -3U to function efficiently ...". This is in conflict with the statement before this sentence which states that the -3 position of 3' ss can be either U or C (instead of strictly requiring U).

7. Page 12, in Discussion. I believe wobble base pairing typically refers to the third base of the codon. More accurately speaking, the 3' ss is recognized through non-Watson Crick basepairing as reviewed by multiple yeast and human spliceosomal P complex structures. The authors should also cite the following yeast P complex structure papers: PMID 29153833 and 29146870.

The authors stated that "However, in early spliceosomal complexes, A or B complex, the 3' ss could not be identified because the electron density showed disorder (32, 33)". This is largely true for the human spliceosome structure (ref 33), but ref 33 is the structure of pre-B not the A or B complex. The authors should correct this. The authors should also remove ref 32 which is a yeast spliceosome structure - In yeast, 3' ss is not recognized in early stages of the splicing cycle, not because the density in the structure is poor.

8. The paper will benefit from a careful proofreading by a native English speaker to correct many wording and grammar errors.

Subject: Nature communications manuscript #NCOMMS-19-40870 (Elucidation of the aberrant 3' splice site selection by cancer-associated mutations on the U2AF1)

We wish to thank reviewers for leading to improve the manuscript. Our detailed responses to the reviewers are given below. And we have re-analyzed crystal structures with the non-crystallographic symmetry restraint, in order to answer the reviewer's comment. Our structures have 9 molecules in the asymmetric unit, and use of NCS restraint gives much more reliability. We have deposited new structure coordinates in PDB.

Best wishes,
Eiji Obayashi

Reviewer 1

Major comments:

1. The studies focus on the fission yeast homolog rather than the disease-relevant human homolog. How similar are the proteins? Please justify extrapolating the results from the yeast homolog to interpretations of the human disease.

I also feel that the use of the yeast protein should be more clear and honestly disclosed in the manuscript text.

We have included a new paragraph discussing how similar the yeast and human U2AF1 are (Page 4 and 5), and a supplementary figure (Supplementary Fig. 1). The yeast U2AF1 has high sequence identity with the human one, 60%, except for the RS domain (Supplementary Fig. 1). Further, amino acids involved in the binding to RNA elucidated in this study, and also the pathogenic hot spot, Ser34 and Asn157, are all conserved. The structure of yeast U2AF1, therefore, can be a good model for human U2AF1 in order to know how the mutation on U2AF1 leads to diseases.

2. The S34Y mutation is minor in MDS compared with the major S34F mutation. Please clarify why the S34Y mutation chosen as the focus of structure determination,

how common it is in MDS, and how an S34F mutation might be similar in effects or differ.

We have solved the structure of S34Y mutant U2AF1 instead of S34F mutant because the solubility of S34Y is higher than that of S34F. Indeed, we have tried to crystallize the S34F mutant as well, but unfortunately could not manage it. We mentioned that reason in Page 11. For other assays, we have used both S34Y and S34F and effects of mutation looks very similar. In MDS, S34F is the most common mutation (66%) and S34Y mutation is the one-sixth (11%).

3. The Q157P/R mutations are the next most common in MDS after S34F. Please describe the Q157 homologous residue of the fission yeast protein. How do the Q157 mutations affect splicing and is it supported by RNA interactions? This information is available from the structure but does not seem to have been analyzed.

We have included a new paragraph discussing the mutation effect of another hot spot in U2AF1 for disease, Q157 in Page14. Q157, is conserved in *S. pombe* U2AF1 at 151. In the structure of yeast U2AF1, Q151 is located between -1G and +1G bases, so mutation at this position could affect to the base recognition.

4. The S34F mutation of the human protein influences -3C vs -3U choice rather than -3Py vs -3A. The focus on the -3A for structural biophysics seems misplaced or peculiar to the yeast protein. Can the structures give insight into why the human protein splices -3C and skips -3U?

Several studies showed that the mutation at Ser34 of U2AF1 alters 3' splice site preference. Expression of S34F/Y mutant U2AF1 promotes recognition of the 3' splice site bearing a C or A immediately preceding the AG (CAG or AAG). Although the selection of CAG sequence for 3' splice site by S34F/Y mutant is major, especially in lung adenocarcinoma, the appearance of A at -3 position in MDS reported by Iligan et al. cannot be ignored. Therefore, we discussed effects of mutation at Ser34 against both CAG and AAG sequences in Pages 9-13.

5. The resolution of the structures does not appear sufficient to resolve hydrogen bonds since the Rmerge is >1 in the highest resolution shell. Moreover, the highest reported resolution is 3.35 for the S34Y structure. Please report Rpim. To what resolution is the Rpim $<50\%$ for the three structures, i.e. reliable?

There has been an extensive discussion recently on the CCP4 bulletin board regarding exactly this question, and whether it is appropriate to throw away data beyond some artificial limit. The consensus is to keep as much data as possible. We have now reported Rpim in Table 2. Although the resolution is not high, the presence of 9-fold symmetry and use of NCS gives greater confidence in the model than the headline resolution number.

Especially since only five main text figures are planned and the focus of the paper is crystal structures, the experimental data table would be more appropriate given in the main text, for full disclosure of the limits of the structure quality to the reader.

As mentioned by reviewer, we show the experimental data table for crystallographic study in the main text as Table 2.

6. Please state the coordinate error and if justified by the structure quality report hydrogen bond distances. If reporting hydrogen bond distances cannot be justified, please say so for the non-structural biologist audience.

The coordinate error is reported by PHENIX to be 0.45 for WT-UUAGGU, 0.53 for WT-UAAGGU and 0.49 for S34Y-UUAGGU. As noted above, however, we feel the unusually high degree of NCS increases the reliability of the model. We note that it is common for B factors in crystal structures to imply mean displacements easily comparable with oxygen-hydrogen distances.

In a few cases, the interactions appear to be mis-interpreted. For example, cysteines that are liganded to zincs are said to be involved in hydrogen bonds. This is highly unlikely, especially since cysteine side chains are hydrophobic in character. I

suggest rephrasing these statements.

Ligplot could be used to plot the interactions in an unbiased manner.

The Cys coordinating zinc ions are in hydrophobic environment (in agreement with the reviewer's comments), and the character of the sulfur atom could be similar to that in methionine or in half cysteine, as suggested in the paper (PMID: 29230465, Kluska et al. 2018). However, it can act as proton acceptor when a suitable proton donor is nearby, since the sulfur atom is fairly electronegative. In the present structure, N6-amino group of the base (-2A) is close enough to interact to the sulfur atom of Cys27, the distance between 3.07 and 3.35 Å. The LIGPLOT also indicates this interaction showing in Supplementary Fig. 11. Furthermore, papers reporting the structure of other CCCH-type zinc finger show hydrogen bonds between the sulfur of cysteine coordinating to zinc atom and the amino group of the RNA base (PMID: 24071581, Kuhlmann et al. 2014). In line with the literature, we suggest a hydrogen bond between Cys27 and N6-amino group of -2A.

7. Please give an unbiased electron density map of the RNA interactions. The 2Fo-Fc map shown in Supplementary data is a biased map and should not be used for publication.

Following the referee's comment, we have replaced the *2Fo-Fc* map to the *mFo-DFc* omit map of the RNA, shown in supplementary Fig. 10. The standard difference map in the submitted manuscript has been created by *Polder Maps* to reduce bias.

8. Table of ITC data are given, but the data lacks errors and statistical significance. Were the experiments replicated? If not, the data should be replicated before publication. Please provide standard deviations of replicates and t-tests. Also, ALL isotherms should be shown to support the quality of the data.

We have replicated ITC experiments and have provided standard deviations in Table

1. And all data measured by ITC experiments are shown in Supplementary Fig. 12 and Supplementary Table 1.

9. Appropriate literature in the field is mis-cited (or not cited). Examples include but are not limited to:

“misregulation underlies many human diseases 3-7.” Has mix of primary lit and reviews and all hematologic malignancies not “many human diseases”.

The reference for U2AF binding the intron is a review, which I don't believe even mentions U2AF.

“U2AF2, interacts both with the polypyrimidine tract in the intron and to SF3B1”

Reference 11 does not have anything to do with SF3b binding U2AF. Reference 1 is a review.

U2AF1 contacts 3'SS – 13 & 14 are wrong, it should be Green Nature and Valcarcel MCB

Aberrant splicing due to mis recognition of 3pSS .. “and so on” – please elaborate.

References are reviews, could be primary for MDS, only two major ones Yoshida and Graubert

Ilgan and Kim S34F/Y made splicing program “confused” to enhance exon inclusion

Should be Graubert Leukemia and Varmus Plos Genetics as well as Ilgan

Not just exon inclusion – exon skipping, cryptic 3' SS, alternative 3'SS also occur

Papers mentioned by the referee are now cited in our manuscript.

10. There are only a few structures of CCCH zinc knuckles bound to RNA, including this one. Could the authors please compare the RNA orientations between the two U2AF1 zinc knuckles and also with the other available high resolution structures of zinc knuckles (Unkempt and Yth1 as well as the two cited herein, possibly others). Are there common themes? (an aromatic-arg packing is mentioned but this reviewer is surprised because by memory I think the available structures are very different from one another with regard to orientation of the bound RNA). This information would be useful to the field of RNA binding proteins.

We have included a new paragraph discussing the structural comparison between the present structure and other CCCH-type zinc fingers bound to RNA in Page 14-17 and Supplementary Fig.8 and 9.

Minor comments:

11. U2AF1 does not have an RRM; It has a U2AF Homology Motif (UHM)

We have corrected the wording as suggested.
RRM to U2AF homology motif (UHM)

12. “crystal lattice” – not due to “lattice”, should be due to “packing”

We have corrected the wording as suggested.

13. How many atoms of what type contribute to the reported RMSDs

We have calculated $C\alpha$ rmsd for each U2AF1 molecule in the asymmetric unit, comparing the models with and without bound RNA. We have added the number of atoms used in each calculation in the table, is shown Supplementary Fig. 2b.

14. Please use prime rather than single quote for 3' or 5' of RNA

We have corrected the wording as suggested.
5', 3' splice site to 5', 3' splice site

Reviewer 2

1. Page 7. There are multiple places where Fig. 4a was cited while Fig. 3a seems to be the correct figure.

We have corrected the figure references.

2. Page 7. The authors stated that “It was believed that Phe20 only contacted the distal

region of the base, preventing the steric hindrance with the sugar backbone, and favouring purine over smaller pyrimidine base at this site.”

Who believed? And what is evidence that support this belief?

In the crystal structure of U2AF1 complexed with RNA, Phe20 only contacted the distal region of the base of -2A. Since the position of Phe20 in the RNA-bound form is almost the same as that in the RNA-free form (Supplementary Fig. 5), Phe20 hardly moves from this position. The base position of -2A is fixed by steric interactions with the sugar backbone. Therefore, it seems difficult that smaller pyrimidine interacts to Phe20 (Supplementary Fig. 5). In other words, purine is favored over smaller pyrimidine bases at this site. We mentioned this in Page8.

3. Page 7. I am not convinced that the authors' argument that Cys27 strictly discriminates the adenine base is sound. The authors stated that Cys27 forms a hydrogen bond with the N6-amino group of -2A. If A were to be changed to G, the corresponding 6th position will be a carbonyl oxygen which will work as a proton acceptor and cannot form the equivalent hydrogen bond. However, why can't the sulfur atom of Cys27 serve as a hydrogen bond donor in this case to still form a hydrogen bond with a G nucleotide at -2 position? In fact, there are literature (for example PMID 19089987) reporting that the sulfur atom of a Cys is a poor hydrogen bond acceptor, but moderately good donor.

The paper which reviewer indicated (PMID: 19089987, Zhou *et al.*, 2008) showed that cysteine residue is a very poor H-bond acceptor but a moderately good H-bond donor. In our case, however, Cys27 is coordinating to zinc ion. The cysteine residue coordinating to zinc ion should be deprotonated (PMID: 29230465, Kluska *et al.* 2018). So, the character of the sulfur atom could be similar to that in methionine or in half cysteine showing in the paper, Zhou *et al.* Since the sulfur atom in cysteine residue has a large electronegativity, it can be a proton acceptor. In the present structure, N6-amino group of the base (-2A) is close enough to interact to the sulfur atom of Cys27, the distance 3.07-3.39Å. Also, the distance is 3.01-3.36Å between Cys163 and N1, and is 3.19-3.52Å between Cys149 and N2 of -1G base. Furthermore, papers reporting the structure of other CCCH-type zinc

finger shows the hydrogen bond between the sulfur of cysteine coordinating to zinc atom and the amino group of the RNA base (PMID: 24071581, Kuhlmann et al. 2014). Therefore, we defined the hydrogen bond between Cys27 and N6-amino group of -2A, and Cys27 may not have this interaction when the base at -2 position is changed from adenine to guanine.

4. The authors stated that the K_d is 46.9 μ M. The binding was never saturated and an accurate K_d cannot be derived from the data the authors presented here.

As reviewer2 said, the accurate calculation of K_d value requires data of which binding is saturated. Some of data in the present study were not saturated in the condition (protein/RNA concentration), especially the case when the binding affinity was low. It can be, however, estimated the binding affinity from such unsaturated data by the curve fitting program. So, we showed calculated K_d values with errors in Figure 3c and Table 1. Also, we showed all low data in Supplementary Fig. 12 and Supplementary Table 1.

5. Page 8. The authors talked about Arg164. Shouldn't residue 164 be an Asn according to Fig. 2b?

We have corrected the residue name.

6. At the beginning of page 10, the authors stated that "These results suggested that while the wild-type protein requires -3U to function efficiently ...". This is in conflict with the statement before this sentence which states that the -3 position of 3' ss can be either U or C (instead of strictly requiring U).

-3 position of 3' SS is occupied by U, but not strictly. Sometimes C is found at this position so have corrected the sentence in Page10 as follows:

"although usually -3 position of 3' SS is occupied by U."

7. Page 12, in Discussion. I believe wobble base pairing typically refers to the third base of the codon. More accurately speaking, the 3' ss is recognized through

non-Watson Crick basepairing as reviewed by multiple yeast and human spliceosomal P complex structures. The authors should also cite the following yeast P complex structure papers: PMID 29153833 and 29146870.

The authors stated that “However, in early spliceosomal complexes, A or B complex, the 3' ss could not be identified because the electron density showed disorder (32, 33)”. This is largely true for the human spliceosome structure (ref 33), but ref 33 is the structure of pre-B not the A or B complex. The authors should correct this. The authors should also remove ref 32 which is a yeast spliceosome structure - In yeast, 3' ss is not recognized in early stages of the splicing cycle, not because the density in the structure is poor.

We have corrected the paragraph in Discussion and have cited papers following by reviewer's suggestion.

8. The paper will benefit from a careful proofreading by a native English speaker to correct many wording and grammar errors.

The manuscript has been corrected by a native English speaker.

REVIEWER COMMENTS

Reviewer #1 (Remarks to the Author):

Yoshida et al. have made a good faith effort to incorporate previous suggestions. Many concerns have been addressed. A few concerns linger or have been raised through the revisions:

Prior Comment #4: The authors have endeavored to include the known preference of S34F mutant U2AF1 for -3C over -3U, but the wording and references could be clarified and expanded as follows. On Pg. 9, "Usually the -3 position of 3' SS is occupied by uridine¹²" Reference 12 does not investigate natural splice site compositions. Rather, I believe the authors could be referring to the SELEX experiment for the U2AF heterodimer. Could the authors please reword this statement for clarity or check the reference?

Likewise, in their response to Reviewer #2, the authors imply that the -3 position is usually occupied by uridine. I am not aware of any evidence to support this belief that human splice sites are usually preceded by -3U. The authors should please check and give appropriate references to support their statements.

Also on Pg. 9, "Ilgan et al. and Kim et al. showed that A or C are found much more frequently at the -3 position of the 3' SS 17,18". It is not clear why these two references were cherry-picked since multiple groups have established the -3C/U preference. For example, Okeyo-Owuor et al. (2015) Leukemia showed the -3C preference of the S34F/Y mutation for MDS-relevant samples at about the same time as Ilgan et al.; Fei et al. (2016) PloS Genetics showed the same for lung cancers before Kim et al. These and possibly others should be included.

"Esfahani et al. reported that in lung adenocarcinomas the U2AF1 S34F mutant preferentially binds to CAG at 3' SS unlike wild-type U2AF1²¹" This statement would benefit from use of more specific language and inclusion of additional references. Esfahani et al. (2019) used iCLIP to show that S34F U2AF1 preferentially crosslinks with CAG splice sites. Okeyo-Owuor et al. (2015) and Fei et al. (2016) first showed that human U2AF1 binds CAG splice sites with higher affinity in quantitative experiments with purified human proteins.

Especially since the RNA binding results here are not entirely consistent with those for the human ternary U2AF1+U2AF2+SF1 complex in Okeyo-Owuor et al. (2015) and Fei et al. (2016) PloS, potential differences between the specificities of yeast and human U2AF complexes also should be acknowledged. Moreover, the ITC experiments here are limited to the minimal U2AF1 heterodimer do not include the physiological situation of the U2AF2 RNA binding domains. A new manuscript (Warnasooriya Nucleic Acids Research) shows that the S34F mutation of U2AF1 influences U2AF2 conformation in a manner that is responsive to the -3 position of the splice site. The possibility that U2AF2 contributes to -3 specificity also should be acknowledged.

Prior Comment #8: Concerns with the ITC data remain.

As also expressed by Reviewer #2, what are the c-values of the isotherms? A c-value outside the recommended limits of approximately 1 – 100 does not give a reliable fit since the curve is flat or a step function. This rule is independent of apparent standard deviations.

Were the isotherms corrected for heats of dilution? Typically, the last saturated portion of the binding curve is subtracted, but in many cases this approach is not possible since the isotherms fail to saturate. What are the n values, which reflect the activity of the macromolecules? Minor comment: The number of replicates in footnote a should be stated as such, not as n=3, because n is the typical nomenclature for apparent stoichiometry of the fitted ITC isotherm.

Lastly, statistically t-tests to demonstrate significance of apparent difference are needed.

Minor comment: The authors also should check the significant digits of the Table 1 ITC results.

Prior Comment #10: The detailed analysis of CCCH zinc knuckles is laudatory and interesting. However, it is my opinion that the new section, which is currently two pages, would benefit from shortening to more concisely present key conclusions.

Minor comment: For the label on Supplementary Fig. 9, the last digit of the Nab2 ZF5 is 4 L J 0 (zero) rather than the letter O.

Reviewer #2 (Remarks to the Author):

I feel the explanations the authors provided in response to item 3 should be incorporated in the manuscript. The authors have otherwise addressed my concerns.

Subject: Nature communications manuscript #NCOMMS-19-40870 (Elucidation of the aberrant 3' splice site selection by cancer-associated mutations on the U2AF1)

We wish to thank reviewers for leading to improve the manuscript. Our detailed responses to the reviewers are given below.

Best wishes,
Eiji Obayashi

Reviewer 1

Major comments:

1. The authors have endeavored to include the known preference of S34F mutant U2AF1 for -3C over -3U, but the wording and references could be clarified and expanded as follows.

On Pg. 9, “Usually the -3 position of 3'SS is occupied by uridine¹²” Reference 12 does not investigate natural splice site compositions. Rather, I believe the authors could be referring to the SELEX experiment for the U2AF heterodimer. Could the authors please reword this statement for clarity or check the reference? Likewise, in their response to Reviewer #2, the authors imply that the -3 position is usually occupied by uridine. I am not aware of any evidence to support this belief that human splice sites are usually preceded by -3U. The authors should please check and give appropriate references to support their statements.

As mentioned by reviewer, we checked the -3 position at 3'SS in mammals and cited the following reference in Page 9;

Sheth, N. *et al.* Comprehensive splice-site analysis using comparative genomics. *Nucleic Acids Res.* **34**, 3955–3967 (2006).

Also, we mentioned the sequence preference of U2AF complex elucidated by SELEX experiment with reference (Wu et al. Nature 1999) as a kind suggestion by reviewer.

2. Also on Pg. 9, “Ilagan et al. and Kim et al. showed that A or C are found much more frequently at the -3 position of the 3'SS 17,18”. It is not clear why these two references were cherry-picked since multiple groups have established the -3C/U preference. For example, Okeyo-Owuor et al. (2015) Leukemia showed the -3C preference of the S34F/Y mutation for MDS-relevant samples at about the same time

as Ilagan et al.; Fei et al. (2016) PloS Genetics showed the same for lung cancers before Kim et al. These and possibly others should be included.

“Esfahani et al. reported that in lung adenocarcinomas the U2AF1 S34F mutant preferentially binds to CAG at 3’SS unlike wild-type U2AF121” This statement would benefit from use of more specific language and inclusion of additional references. Esfahani et al. (2019) used iCLIP to show that S34F U2AF1 preferentially crosslinks with CAG splice sites. Okeyo-Owuor et al. (2015) and Fei et al. (2016) first showed that human U2AF1 binds CAG splice sites with higher affinity in quantitative experiments with purified human proteins.

The papers mentioned by the reviewer (Okeyo-Owuor *et al.* (2015) Leukemia and Fei *et al.* (2016) PloS Genetics) are now cited in our manuscript in Page 4 and 9.

3. Especially since the RNA binding results here are not entirely consistent with those for the human ternary U2AF1+U2AF2+SF1 complex in Okeyo-Owuor et al. (2015) and Fei et al. (2016) PloS, potential differences between the specificities of yeast and human U2AF complexes also should be acknowledged. Moreover, the ITC experiments here are limited to the minimal U2AF1 heterodimer do not include the physiological situation of the U2AF2 RNA binding domains. A new manuscript (Warnasooriya Nucleic Acids Research) shows that the S34F mutation of U2AF1 influences U2AF2 conformation in a manner that is responsive to the -3 position of the splice site. The possibility that U2AF2 contributes to -3 specificity also should be acknowledged.

As mentioned by the reviewer, our results can hardly explain the mechanism of the aberrant splicing especially at CAG 3’ splice site by S34F mutant of U2AF1 as seen in adenocarcinoma, although our data fit match with the splicing at AAG 3’ splice site by S34F/Y mutants of U2AF1. Since CAG 3’SS is well spliced even by WT U2AF1, splicing efficiency for CAG 3’SS by S34F is almost same to that by WT of U2AF1. Therefore, as the reviewer said, it may be necessary to consider including other splicing factors working with U2AF1, U2AF2 and SF1. However, the investigation of the regulation of splicing of U2AF1 with U2AF2, SF1 and other splicing factors is outside of the scope of this manuscript. We have included this in Discussion, Page 16-17, with the reference suggested by the reviewer.

4. Concerns with the ITC data remain.

As also expressed by Reviewer #2, what are the c-values of the isotherms? A c-value outside the recommended limits of approximately 1 – 100 does not give a reliable fit since the curve is flat or a step function. This rule is independent of apparent standard deviations.

Were the isotherms corrected for heats of dilution? Typically, the last saturated portion of the binding curve is subtracted, but in many cases this approach is not possible since the isotherms fail to saturate.

What are the n values, which reflect the activity of the macromolecules? Minor comment: The number of replicates in footnote a should be stated as such, not as n=3, because n is the typical nomenclature for apparent stoichiometry of the fitted ITC isotherm.

Lastly, statistically t-tests to demonstrate significance of apparent difference are needed.

The c-value for the WT-UUm⁶AGGU measurement is 1.01. For each experiment, we have corrected the heat of the injections by subtracting the heat of dilution of the protein into substrate-free buffer. Although the binding curve is not completely saturated, the binding affinity is approximately 100 times lower than for UUAGGU RNA. This is much larger than the expected error in the affinity determination, and therefore we argue that the interaction between Cys27 and N6-amino group of -2A is significant for accurate RNA recognition.

We have performed three independent measurements for all ITC experiments. We have corrected the text of footnote a in Table 1, and we have added the column of stoichiometry (N-value) in Fig. 3, Supplementary Fig. 6 and Supplementary Tables 1 and 2.

5. The authors also should check the significant digits of the Table 1 ITC results.

We have thoroughly checked and corrected Table 1, and ensured each number is quoted to a suitable precision.

6. The detailed analysis of CCCH zinc knuckles is laudatory and interesting. However, it is my opinion that the new section, which is currently two pages, would benefit from shortening to more concisely present key conclusions.

As requested by reviewer, we have shortened the paragraph “Comparison of RNA recognition mechanism by other CCCH-type ZFs”.

7. For the label on Supplementary Fig. 9, the last digit of the Nab2 ZF5 is 4 L J 0 (zero) rather than the letter O.

We have corrected the wording as suggested.

Reviewer 2

1. I feel the explanations the authors provided in response to item 3 should be incorporated in the manuscript.

Since Reviewer 1 is concerned with the length of the manuscript have not incorporated our complete answer in the paper as suggested, but added a brief comment about the zinc ion coordinating Cys27.

REVIEWERS' COMMENTS:

Reviewer #1 (Remarks to the Author):

The authors have addressed this reviewer's concerns. Final constructive corrections include:

- 1) Pg. 16, Discussion: The authors imply that the S34F CAG preference is only observed in lung adenocarcinoma, whereas it also is well-documented in MDS (Okeyo-Owuor et al.; Ilagan et al.). Please reword the Discussion as such.
- 2) The additions to the ITC methods and results add rigor. For transparency, please include that the isotherms were corrected for the average heats-of-dilution from a representative titration of protein into buffer to the Methods and/or Table notes.
- 3) Please round the significant digits of the apparent stoichiometry values in the Supplementary Tables and all figures to the tenths, consistent with the significant digits reported for the thermodynamic values.

Subject: Nature communications manuscript #NCOMMS-19-40870 (Elucidation of the aberrant 3' splice site selection by cancer-associated mutations on the U2AF1)

We wish to thank reviewers for leading to improve the manuscript. Our detailed responses to the reviewers are given below.

Best wishes,
Eiji Obayashi

Reviewer 1

Major comments:

1. Pg. 16, Discussion: The authors imply that the S34F CAG preference is only observed in lung adenocarcinoma, whereas it also is well-documented in MDS (Okeyo-Owuor et al.; Ilagan et al.). Please reword the Discussion as such.

As mentioned by reviewer, we have corrected the sentence in page 16.

2. The additions to the ITC methods and results add rigor. For transparency, please include that the isotherms were corrected for the average heats-of-dilution from a representative titration of protein into buffer to the Methods and/or Table notes.

Following the reviewer's comment, we have mentioned it in Methods.

3. Please round the significant digits of the apparent stoichiometry values in the Supplementary Tables and all figures to the tenths, consistent with the significant digits reported for the thermodynamic values.

As mentioned by reviewer, we have corrected values in Fig. 3c, Supplementary Fig. 6c, Supplementary Tables 1 and 2.